# Assessment of Arctic sea ice simulations in CMIP5 models

# Liping Wu, Xiao-Yi Yang, Jianyu Hu

State Key Laboratory of Marine Environmental Science, College of Ocean and Earth Sciences, Xiamen University, Xiamen, 361102, China *Correspondence to*: Xiao-Yi Yang (xyyang@xmu.edu.cn)

- 5 Abstract. The Arctic sea ice cover has experienced an unprecedented decline since the late 20<sup>th</sup> century. As a result, the feedback of sea ice anomalies to atmospheric circulation has been increasingly evidenced. While the climate models almost consistently reproduce the downward trend of sea ice cover, great dispersion between them still exists. To evaluate the model performance in simulating Arctic sea ice and its potential role in climate change, we constructed a reasonable metric by synthesizing the linear trends and anomalies of the sea ice. We particularly focus on the Barents and Kara seas, where the sea ice anomalies have the greatest potential to feedback the atmosphere. Models can be grouped into three categories according to this criterion. The strong contrast among the multi-model ensemble means in different groups demonstrates the robustness
- and rationality of this method. The potential factors accounting for the different performance of climate models are further explored. The result shows that the model performance depends more on the ozone datasets prescribed by model rather than on the chemistry representation of ozone.

## **1** Introduction

In recent decades, high latitudes in the northern hemisphere turned to show the most visible signals of climate change and surface warming, which is at least twice times of the global average (e.g., IPCC AR5, 2013). This Arctic amplification effect and its mechanism aroused many discussions. The amplification is a consequence of the combination of several factors, among which the retreating sea ice cover plays a central role (Chapman and Walsh, 2007;Serreze et al., 2009;Screen and Simmonds, 2010a). Other important factors contributing to the Arctic amplification of warming include atmospheric and oceanic heat transport, as well as solar radiation force feedback (Holland and Bitz, 2003;Alexeev et al., 2005;Graversen et al., 2008;Serreze et al., 2009;Screen and Simmonds, 2010a;Walsh, 2014).

20 Studies have shown that Arctic amplification of global warming in the last century mainly resided in the lower and middle troposphere, and it was due to the increasing of poleward heat transport by Arctic Oscillation (AO) -- the dominant mode of Northern Hemisphere extra-tropical atmospheric circulation(Graversen et al., 2008). In recent years, with the weakening of atmospheric Arctic Oscillation mode, quadrupole modes of wintertime Arctic sea ice oscillation (over the Pacific and Atlantic sectors of peripheral Arctic Ocean) in the interannual time scale were completely broken down(Yang and Yuan, 2014). Instead, there exists the prevailing decreasing trend of winter Arctic sea ice concentration

(SIC)(Giles et al., 2008). So the changes of sea ice may be a more important factor than the atmospheric internal variability to account for the recent warming trends in the Arctic.

Climate change in the Arctic is asymmetric in different regions (Overland et al., 1997; Venegas and Mysak, 2000; Semenov and Bengtsson, 2003; Rogers et al., 2013). Some places such as the Barents and Kara Seas (BK) are important to the Arctic climate change, probably owing to the

- active exchange of heat and momentum at the ocean-atmosphere interface in these regions. Yang and Yuan (2014) found out that the weakening and collapsing of winter sea-ice double dipole mode can be largely attributed to the thermal effect of significant BK sea ice reduction in late fall and early winter. In addition to the local effect, some regional signal in the atmosphere-sea ice-ocean system can be amplified through a positive feedback and extended to affect the climate in the Arctic as a whole or further in the lower latitudes (Bengtsson et al., 2004;Semenov and Bengtsson, 2003;Semenov, 2008;Semenov and Halford, 2009;Smedsrud et al., 2013).For example, the BK sea ice anomalies may impact the
- European climate through the atmosphere bridge, leading to an unusually cold winter in Europe (Petoukhov and Semenov, 2010). Actually the BK sea ice is widely recognized as an important forcing factor of the mid-to-high latitude wintertime climate such as the position and the strength of storm track, the blocking system, the extreme events and the Arctic Oscillation (Yang et al., 2016;Francis and Vavrus, 2012;Petoukhov and Semenov, 2010;Ruggieri et al., 2016). The distinction of BK sea ice in the Arctic climate system is further reported by Kim et al. (2016). Unlike the any other Arctic marginal seas, the BK sea ice anomalies can extend throughout the year due to the so-called "insulation feedback"
- mechanism. The increased reception of insolation in summer and the heat stored in the BK seas can be released in fall-winter season, resulting in the "delayed warming" in the Arctic (Francis et al., 2009;Deser et al., 2010;Screen and Simmonds, 2010b). This may account for the high correlation between the winter sea ice extent (SIE) in Barents Sea and the summer Arctic SIE in both model simulations and observations (Li et al., 2017). Therefore, it is plausible to highlight the BK sea ice simulation in our assessment of the model performance in casting the Arctic climate change.
- In view of the fact that there are complex interactions between atmospheric circulation and Arctic sea ice, scientists usually use oceanatmosphere-sea ice coupled climate models to diagnose the factors that affect Arctic climate changes and to predict future climate changes. However, the simulation results vary from model to model, because the different grid resolution, initial condition, physical process in the ocean or atmospheric condition can accumulate and lead to a large bias. For example, Turner et al. (2009) using the models of IPCC Fourth Assessment Report (AR4) suggested that stratospheric ozone depletion leads to the observed Southern Ocean sea ice increasing in spring. But model
- experiments by Sigmond and Fyfe (2010) defied the results of Turner et al. (2009) that in response to the stratospheric ozone depletion, the Antarctic sea ice decreases throughout the year, and they asserted that there must be factors other than stratospheric ozone to account for the observed Antarctic sea ice increasing.

Models participating in the Coupled Model Intercomparison Project (CMIP) of the World Climate Research Programme (WCRP) consist of a series of contemporary ocean- atmosphere coupled climate models, by which we can assess whether the models can simulate the trends and anomalies of sea ice correctly. The IPCC fifth assessment report shows that the models participating in the Coupled Model Intercomparison Project phase 5 (CMIP5, http://pcmdi3.llnl.gov/esgcet) have a higher performance than those in the Coupled Model Intercomparison Project

- phase 3 (CMIP3) in their capability of reproducing the long-term trends of sea ice. Rosenblum and Eisenman (2016) found that the inclusion of volcanic activity, rather than improvement of sea ice physics or model resolution, accounts for the priority of the CMIP5 over the CMIP3 in simulating the Arctic sea ice trends. Nevertheless, the CMIP5 model simulation results are far from satisfying, especially in the Antarctic region. Few CMIP5 models capture the observed slight increase trend of Antarctic SIE (Turner et al., 2013;Polvani and Smith, 2013;Zunz et al., 2013;Mahlstein et al., 2013). In comparison, CMIP5 models seem much better in the Arctic sea ice simulation. Stroeve et al. (2012) found that
- the observed seasonal cycle and long-term trend of Arctic SIE were well presented in CMIP5 models. But they noted that the dispersion of projected SIE through the 21st century in CMIP5 models remains similar to that in CMIP3 models. Massonnet et al. (2012) provided several important metrics to constrain the projections of summer Arctic sea ice. Liu et al. (2013) pointed out that by reducing the inter-model spread of the CMIP5 projections, they could reproduce consistent Arctic ice-free time.
- It seems that using models to predict the rate of summer sea ice loss remain uncertain, and results are widely spread among the models. Moreover, the wintertime sea ice decline is remarkable in recent decades, and plays a more important role in driving the climate change in mid-to-high latitudes than the summer sea ice loss. In particular, the sea ice variation in the BK region can bring about powerful feedbacks governing the atmospheric circulation realignment over the northern continents and Polar Ocean, which is projected on the AO mode. (Petoukhov and Semenov, 2010;Alexander et al., 2004;Deser et al., 2004). Although the CMIP5 models can generally capture the AO pattern, there are significant biases in both the magnitude and location of the AO simulation (Jin-Qing et al., 2013). In this study, 30 CMIP5 models are evaluated objectively and
- comprehensively for their capability of simulating the Arctic sea ice variability. Given the strong feedback of BK seas to the Arctic climate and teleconnection with mid-latitude climate through AO mode, we differentiated the BK region from the other Arctic regions (exBK) and endowed it a larger weight. For these two regions, both the long-term trend and the anomalies of sea ice are taken into consideration. In addition to the BK-priority weighting method, we constructed a comprehensive and objective assessment framework to quantify the models' ability of sea ice simulation. Based on this framework, we can sort out some better models to constrain the biases of models and set a better basis for the study of
- Arctic ocean-ice-atmosphere interaction and future Arctic climate change prediction. Moreover, we further scrutinized on the model parameters and suggested the possible way to improve models' performance on Arctic sea ice simulation.

# 2 Data and method

The CMIP5 (Taylor et al., 2012) model simulation dataset can be directly downloaded via the website http://pcmdi3.llnl.gov/esgcet/home.htm. Among others, 30 models are selected for their intactness and availability of sea ice dataset, as shown in Table 1. The HadISST1 SIC data is applied in this study as observation data to evaluate the models (Rayner et al., 2003). HadISST1 is a global monthly SST and sea ice dataset with

- a 1°×1° grid ranging from 1871 to the present, which are taken from a variety of sources including digitized sea ice charts and passive microwave retrievals. The HadISST1 sea ice data are made more homogeneous by compensating satellite microwave-based sea ice concentrations for the impact of surface melt effects on retrievals in the Arctic and for algorithm deficiencies in the Antarctic, and by allying the historical in situ concentrations consistent with the satellite data. As the in-situ data prior to satellite era are sparse and highly inhomogeneous, we truncated both the model and observational data from 1979 afterwards (Rayner et al., 2003).
- We analysed results of historical simulations from 1979 to 2005, prolonged with Representative Concentration Pathways (RCP) 8.5 simulations from 2006 to 2014, except for HadGEM2-CC and HadGEM2-ES with historical simulations from 1979 to 2004 prolonged with RCP8.5 simulations from 2005 to 2014. Given the inconsistency of the various grids and projections in CMIP5 models, we pre-processed the model outputs by interpolating them onto the same grid as HadISST1 data.
- As we underlines the significance of BK sea ice variability in recent climate change, we applied a weighted scoring method. The detailed 15 processes of quantification are as follows. Firstly, we multiply the fraction of grid cell covered by sea ice (SIC) to the area of grid cell to calculate the sea ice area (SIA) for the exBK and BK regions, respectively(formulation shown as Eq. (1)).  $SIA = \sum_{lon=-179.5}^{179.5} \sum_{lat=89}^{lat=89} SIC(lon, lat) * 2\pi * r^2 * (sin(\frac{lat+1}{180} * \pi) - sin(\frac{lat}{180} * \pi))/360$  (1)

Then their linear trends are estimated using the least square method. Comparing the SIA trends of models output with observations, we calculate the relative errors of the trend (Eq. (2)). The lower the absolute value of relative error is, the better the performance of models.

relative error =
$$\left|\frac{X_{mod} - X_{obs}}{X_{obs}}\right|$$
 (2)

Secondly, we obtain the detrended SIC anomaly time series in each grid, with both the climatology and the linear trend being subtracted from the original data. A quantitative comparison between the model results and observations is conducted using the method developed by Warner et al. (2005) (Eq. (3)),

$$Skill = 1 - \frac{\sum_{i=1}^{N} |x_{mod} - x_{obs}|^2}{\sum_{i=1}^{N} (|x_{mod} - \bar{x}_{obs}| + |x_{obs} - \bar{x}_{obs}|)^2}$$
(3)

where X is the variable,  $(\bar{X})$  is its time mean, and the subscripts mod and obstand for model results and observations, respectively.

We calculate the skill values (Sk) in each grid, and then we get a final skill value after averaging them. It is clear that the higher the score, the better models' performance is.

Finally, we use a weighting average method to collaborate the trend error (Eq. (2)) and anomaly skill (Eq. (3)) quantitatively. As the BK sea ice may exert a far-reaching effect on the Arctic climate and remote effect on the Northern Hemisphere extratropical atmospheric circulation, we

weighted it more. So the weight coefficients of four factors including sea ice trends of exBK and the BK, sea ice anomalies of the exBK and the BK are 0.1, 0.3, 0.2 and 0.4, respectively. To bridge the gap between the relative error and the skill score, we used the residual relative error (RRE) (one minus the absolute value of relative error) instead of relative error itself. We then get the final score by normalizing the raw scores and multiplying the weighted coefficient (Eq. (4)).

score = 0.1 \* exBKRRE + 0.2 \* exBKSk + 0.3 \* BKRRE + 0.4 \* BKSk

(4)

The final scores for 30 models are listed in Table 2.

## **3 Results**

As is noted above, we get a quantitative score of each model according to our methods. The 30 models are then divide into three groups based on their weighted scores. Models with the score>1 as group I (high score model group), score<-1 as group III (low score model group) and others as group II (medium score model group) (Table 2). According to this scoring criterion, 4 model members in group I, namely MPI-ESM-MR, MPI-ESM-LR, NorESM1-M and GFDL-CM3 exhibit the best performance in Arctic sea ice simulation. In contrast, 3 model members in group III, namely FGOALS-g2, MIROC-ESM and CSIRO-Mk3-6-0 are the lowest scoring models. The 23 members in group II, including the majority of CMIP5 models, are labelled as in the medium level of Arctic sea ice simulation ability. It is evident that the number of model members in each

group depends on the subjective choice of score cut-point. But our results in this study are not quite sensitive to the number of group members. The significance discrepancies between the group multi-model ensample means are still visible even if the ±0.8 scores are adopted as the basis of grouping instead of ±1.0 (figures not shown). To verify and validate our methods, we select several metrics to check the rationality and robustness of our scoring and grouping system.

#### 3.1 Multi-model ensemble mean state

## 3.1.1 Sea ice area

Figure 1 exhibits the climatological seasonal cycle of SIA in each group. The root mean square errors between the multi-model ensemble mean (MMEM) of each group and observation are 0.37, 0.65, and 1.2 million km<sup>2</sup> for groups I, II and III, respectively. Apparently, MMEM of group

I is more conformable to observation than that of either group II or group III. Observed SIA reaches maximum in March and minimum in September. Models in group I seem consistently well reproducing this feature. But for group II and III, the dispersions among model members are obviously enlarged. Though almost all of the models qualitatively simulate the seasonal wax and wane of sea ice, the values of monthly climatology of SIA vary as far as 5-10 million km<sup>2</sup> apart in groups II and III. Even the MMEMs are obviously overestimated in wintertime for group II, and underestimated in winter and overestimated in summer months respectively for group III. To further probe the spatial distribution of this SIA climatology, we calculate the SIA in each longitude and present the Meridional climatological sea ice area in March and September for each group (Fig. 2). In March (Fig.2a), MMEM sea ice areas of all the three groups appear highly conformable with the observations in longitudes from 60° E to 140° E and 122° W to 180° W, including Kara, Laptev, Bering and Beaufort Seas. In other longitudes, Group I behaves much better than the other two groups in general. Group II MMEM tends to overestimate the sea ice area in Baffin Bay, Greenland, Barents and Norwegian Seas, whereas Group III MMEM tends to underestimate (overestimate) the sea ice area in Hudson (Barents) and adjacent regions. Wider gaps among the three groups exists in September (Fig.2b) than in March. Group III models overestimate the sea ice area in most part of Arctic oceans, with the poorest performance in Barents and Kara Seas. The overall skills between the MMEM of each group II over Groups II and III is most striking in the BK region in March and September, which is foreseeable with the larger weights given to the BK in our evaluation system.

## 15 3.1.2 Trend of sea ice area

The SIA trend is also estimated for each longitude (shown in Fig. 3). All MMEMs of SIA trend show a decreasing trend everywhere, which qualitatively agree with observation results except in the region near Greenland where the observed SIA is increasing. The MMEMs of SIA trend of 3 groups are well distinguished in BK. The MMEMs of group I and II are much closer to observation than that of group III. In other regions, the differences between three MMEMs and observation are less significant. The skills between the MMEM of each group and observation are 0.77, 0.74 and 0.64 respectively.

#### 3.1.3 Sea ice variability

The skill score distributions of MMEM detrended SIC anomaly for each group in exBK region and BK region are displayed in Fig. 4 and Fig. 5, respectively. In the BK, the skill scores of group I are much higher than that of group II and groupIII everywhere (Fig. 5). As for the exBK region, the differences of three groups are asymmetric and less significant. The most rigid hierarchy is located in the Beaufort Sea and Laptev Sea

among the three groups. Generally speaking, the mean of spatial MMEM of skill scores for group I, II and III are 0.28, 0.25, 0.23 in exBK and 0.38, 0.29 and 0.19 in BK, showing an obvious descending in the sequence of group number both in the BK and exBK. It seems that the models

performing a better simulation of sea ice in BK region may generally acquire higher scores in other Arctic regions too. The exceptions appear in the Greenland Sea, the Baffin bay, the Bering and Okhotsk seas, where the skill scores of group III are almost as high as group I and the group II get the lowest score (Fig. 4). This is probably due to the larger dispersion between the model members in group II as its model number is much larger than the other two groups.

## 5 3.2 Individual model contrast

As is shown, the remarkable contrast among the multi-model ensemble means in different groups proves that the weighted method is applicable to distinguish the models' capability of sea ice simulation in general. However, the ensemble means blur the individual differences between the group members. It is quite necessary to further probe the detailed skill score model by model. We thus applied a heatmap to interpret the model differences in various terms of skill scores (Fig. 6). The heatmap is a graphical representation of data where the individual values contained in a

- matrix are represented by colours (https://en.wikipedia.org/wiki/Heat\_map). In our two-dimensional heatmap, the models are arranged in descending order of total skill score in the x-axis, superposed by scores of sub-item in the y-axis. The colour squares are the normalized score values for each model and each sub-item. It is apparent that the first 4 models in group I generally achieve the much higher scores for each sub-item than the last 3 models in group III. The models in group II exhibit a rather chaotic order in the sub-item scores, which to some extent reflects their large dispersion. The contrast between these groups is more pronounced for BK anomaly than for exBK trend, as we assigned different
- weights to these sub-items. It is noteworthy that the score of exBK trend for the 4<sup>th</sup> model (model No.13 GFDL-CM3, group I) is extraordinarily low (≤-1.0), in great contrast to the extraordinary high score (≥1.0) of the 29<sup>th</sup> model (model No. 22 MIROC-ESM, group III). Moreover, the scores of exBK anomaly of these two models are very close (0.83vs 0.80), which is out of our expectation and should be further investigated. To figure it out, we present a detailed comparison of sea ice simulation of the two models with the observations (Fig. 7). Figure 7a shows the annual mean SIA time series. No.22 and No.13 models exhibit a similar downward trend, which is consistent with observation. But No.13 model
- overestimates the ice decline trend to some extent, particularly for recent two decades. No. 22 model, though reproducing the observational sea ice decrease rate, underestimates the climatological SIA dramatically and the linear correlation coefficient of No.22 with the observation data is 0.57, smaller than that of No.13 (0.77). To compare these two model simulation results in detail, we present 10-year running mean of detrended exBK SIA skill scores for both two model simulations and observation (Fig. 7b). Model No.13 seems better than model No.22 in simulating SIA variability prior to 2007.In recent decade, however, the two models perform roughly the same. As for the simulation of detrended exBK SIA
- anomaly on spatial scale, we present spatial distribution of skill scores (Figs. 7c and 7d). The SIC skill scores of model No.13 are extremely low around the southern boundary of the ice cover (near zero), which is owing to its inconsistent sea ice edge simulation with observation in winter On the other hand, this model performs well in simulating the SIC in eastern Arctic ocean including Laptev, eastern Siberian and ChukChi Seas.

The scores of model No.22 are still low in the ice edge area and high in the central Arctic Ocean. But the contrast between the high and low score values is not so great as model No.13. From the comparison between the two models, it is obvious that model No.13 is still a better choice than model No. 22 in simulating the sea ice variability.

## 3.3 Potential factors accounting for the different performance of climate models

- According to the results of the analysis above, the weighted score can well measure the model's capability to simulate the sea ice. But the reason for the dispersion of model simulations, particularly in group II, remained unknown yet. The different parameters of the model itself, the grid resolution, and the way of models to represent stratospheric ozone have been proposed to be the potential factors to affect the model performance in sea ice simulation (Turner et al., 2009;Sigmond and Fyfe, 2010;Zunz et al., 2013). To investigate the ozone effect on the model performance, we listed the ozone representation and the prescribed ozone datasets for 30 CMIP5 models reorganized by Eyring et al. (2013) in Table 3.
- According to ozone representation, these 30 models can be roughly grouped into two categories. One contains 10 models with interactive or semioffline chemistry ozone representation (CHEM, bold-size model names in Table 3), and the other contains 19 models with prescribed ozone representation (NOCHEM, normal-size model names in Table 3). Most of the NOCHEM models apply the prescribed ozone both in stratosphere and in troposphere. The exception is HadGEM2-ES, which uses prescribed ozone in stratosphere but interactive ozone chemistry in troposphere. We thus ruled outHadGEM2-ES in the following statistics to avoid the possible ambiguity. Semi-offline is denoted if the prescribed ozone dataset
- has been, unlike the models with prescribed ozone, calculated with the underlying CMIP5chemistry-climate model using prescribed SSTs and SICs following historical emissions from Lamarque et al. (2010)and future emissions under the RCP scenarios as described by Lamarque et al. (2011). They differ from the class of prescribed ozone CMIP5 models (NOCHEM), because their stratospheric ozone evolution responds to changes in GHG concentrations in the four RCPs, although it is still calculated offline(Eyring et al., 2013). The average scores in four metrics and their weighted mean are calculated respectively and displayed in Fig. 8a. For the weighted mean, the scores are -0.22 for NOCHEM models
- and 0.41 for CHEM models. The strong contrast of scores between the two categories suggests the superiority of interactive or semi-offline ozone chemistry over the prescribed ozone representation. By checking model score groups in Table 2 and ozone representations in Table 3, we found that 8 out of the bottom 10 models applied prescribed ozone representation (NOCHEM). Nevertheless, the CHEM and NOCHEM models are well-matched in the top group (2 CHEM models vs. 2 NOCHEM models in group I and 6 CHEM models vs. 4 NOCHEM models in top 10 models). The sea ice anomaly skill scores are well consistent with the weighted mean scores. The linear trend scores, on the contrary, are not so
- sensitive to the choice of ozone representation. This suggests that to apply the interactive and semi-offline ozone chemistry may greatly improve the model performance of Arctic sea ice simulation in interannual to decadal time scales, but have little effect on the linear trend simulation.

In order to further explore the effect of ozone on model performance, we subdivide the models with prescribed ozone representation and semi-

offline ozone chemistry according to the ozone datasets they used. Here we examine whether various ozone datasets affect the simulation of sea ice. The ozone datasets used by 18 NOCHEM models include C<sup>1</sup> (10 models), P<sup>5</sup> (2 models), P<sup>7</sup> (2 models), C<sup>2</sup><sub>modA</sub> (2 models), C<sup>3</sup><sub>modA</sub> (1 model) and C<sup>4</sup><sub>modB</sub> (1 model). The latter two datasets are excluded here because their sample sizes are too small. The ozone datasets used by semi-offline chemistry models include P<sup>2</sup> (3 models) and P<sup>6</sup> (3 models). We also estimate the scores of the interactive ozone chemistry models (4 models) in comparison with other models though they do not have any prescribed ozone data. The average scores of sub-group models in terms of four metrics (BK trend exBK trend, BK anomaly, and exBK anomaly) and their weighted mean scores are calculated respectively and shown in Fig. 8b. For the weighted mean scores the  $C^2$ , we models (core 1.2) are far superior to the other three categories in the NOCHEM group and even

- 8b. For the weighted mean scores, the C<sup>2</sup><sub>modA</sub> models (score 1.2) are far superior to the other three categories in the NOCHEM group and even better than CHEM models. Moreover, the C<sup>2</sup><sub>modA</sub> models prevail over the other models in that they achieve high and balanced scores in all four metrics. The P<sup>2</sup> models in semi-offline group, in contrast to the P<sup>6</sup> models, achieve the secondary high score of 0.74. Like the interactive ozone models (I), the P<sup>2</sup> models performs well in simulating the sea ice anomalies but unsatisfactorily in simulating the linear trends. The preference of
- models (i), the 1<sup>-</sup> models performs were in simulating the sea fee atomates out distantiation of the simulating the index of the preserved of the models and the bottom 10 models. Both two models with the prescribed  $C^2_{modA}$  ozone data and 2 out of 3 models with the semi-offline P<sup>2</sup> ozone data rank in top 10 models, while none of the models with these two ozone datasets rank in bottom 10 models. It seems that the model performance depends more on the ozone datasets quality than on whether ozone is prescribed or interactive. Although the models show remarkable bias on the choice of ozone datasets, other factors may also
- 15 influence the model preference. For the grid resolution, the potential relationship with four metrics and weighted mean scores are also investigated (figure not shown). The result is ambiguous. The scores of models with high resolution can be higher than those with low gird resolution, and also can be lower, and even models with the same resolution can show a totally different sea ice simulation ability (e.g. model No.27 NorESM1-M and No.28 CESM1-WACCM). However, to exclude the influence of model design itself, we compare two sets of models which have the same sea ice model and the same stratospheric ozone representation but the different spatial resolution (MPI-ESM-LR, MPI-
- 20 ESM-MR and IPSL-CM5A-LR, IPSL-CM5A-MR). The resolution of MPI-ESM-MR (IPSL-CM5A-MR) is higher than that of MPI-ESM-LR (IPSL-CM5A-LR), and the score of high resolution models (MPI-ESM-MR and IPSL-CM5A-MR) are 0.34 and 1.14 higher than that of low resolution models (MPI-ESM-LR and IPSL-CM5A-LR). Nevertheless, it is hard for us to conclude what role the spatial resolution plays in sea ice simulation due to small sample size. Too many other parameters can affect the model's performance. Thus it remains an open question which factor dominates the models' capability of Arctic sea ice simulation.

### 25 4 Summary

5

In this study Arctic sea ice simulation in models that participated in CMIP5 has been evaluated. Four metrics including the long term trends and SIC anomalies in the interannual and decadal time scales in the Barents and Kara seas (BK) as well as the other Arctic regions (exBK) are

integrated with different weighted coefficients. Models are divided into three groups according to the weighted scores. The feasibility and robustness of this assessment method is verified for various items. Finally, some parameters (mainly ozone representation) of model itself are investigated to explain the discrepancies between models.

In general, the multi-model ensemble means (MMEM) of three groups are well distinguished in annual cycle, linear trends as well as interannual variability of sea ice area (SIA), which demonstrates the rationality of our evaluation criterion. For individual model, some high ranking models are superior to low ranking models not in all metrics but in the core metrics like BK trends and anomalies. This underlines the necessity of the weighting method in model assessment and to some extent verifies the results in this study. Previous studies also evaluated the model

- performance on sea ice simulation (Shu et al., 2015;Semenov et al., 2015;Maslowski et al., 2012). Climate models in general reproduce the sea ice retreat trend in the Arctic during the 20th century and simulate further sea ice area loss during the 21st century in response to anthropogenic
  forcing, but these models suffer from large biases and the results exhibit considerable spread (Zhang and Walsh, 2006;Semenov et al., 2017;Arzel et al., 2006). Our results generally agree with them in the linear trend and model dispersion. However, they usually used the multi-model
- ensemble mean without discrimination to abate the model bias. Most assessments pay more attention to the sea ice tendency of MMEM (Shu et al., 2015;Li et al., 2017;Stroeve et al., 2012). However, simulations vary from model to model and the sea ice variability is also an important factor for atmospheric circulation. Our assessment underlines the capability in simulating the BK sea ice trends and anomalies, due to their significance in driving the recent climate change both in the Arctic and in the mid-latitudes for each model. Our evaluation results can be a useful
- reference for model developers to improve the simulation of AO, and it may be a considerable idea to check the sea ice simulation in BK. In addition, we make a preliminary discussion regarding the reasons of the model dispersion. Several parameters of models including resolution and ozone representation are investigated. Results show the model resolution does not significantly impact the model performance, which is in accordance with Rosenblum and Eisenman (2006). Instead, the ozone datasets that models used may be an important factor. It can be set as a
- 20 reference for NOCHEM model developers that using  $C^2_{modA}$  ozone data seems to be a wise decision. Although we emphasized the importance of BK sea ice simulation in our assessment system, it should be regarded as only a necessary but not a sufficient condition for climate models to achieve the goal of predicting future climate change. Further research is required to explore the detailed physical processes and mechanisms through which the ice anomalies exert their influences onto the local and remote climate variability.

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

25 Norwegian earth system model, NorESM1-M-part 2: Climate re