# Peer review of "Assessment of Arctic sea ice simulations in CMIP5 models"

_The Cryosphere, 2018_

## Referee Comment (RC1) · Anonymous Referee #1 · 23 Apr 2018

The study applies a new metric to assess the ability of CMIP5 models to accurately simulate Arctic sea ice evolution. Based on their results, they suggest that model performance strongly depends on ozone datasets and how well the Barents and Kara seas are simulated.

Originality: I very much like the idea of using physical mechanisms to examine the accuracy of Arctic sea ice evolution in climate models.

Scientific Quality: The authors, motivated by a number of studies outlined in the introduction, choose a metric to group the models based on how well they simulate the Barents and Kara seas. However, there choice of metric seemed arbitrary at best, and there was little analysis done to validate this method. For example, perhaps it could be useful to examine the relationship between BK sea ice trends vs pan-Arctic sea ice

trends. This might help to convince readers of the method, and therefore make the rest of the analysis more convincing.

Additionally, I was a bit surprised to not see Stroeve and Notz 2015 referenced in this paper. Stroeve/Notz go into great detail regarding how well Arctic sea ice is simulated by CMIP5 models

Significance: I really like the idea of this sort of analysis, but it was not clear to me how this method improves on earlier studies that examine how well sea ice is simulated in climate models (e.g. Stroeve and Notz 2015). For example, it could be really interesting to examine what is learned by using their metric versus just examining the pan-Arctic, as is often done in these sorts of analyses?

Presentation: There were a number of grammatical errors and typos throughout. Further, the introduction was a bit long, unclear, and took up about a 1/3 of the paper. Also, there seemed to be several references to Antarctic sea ice papers, though this was entirely an analysis of the Arctic sea ice. There wasn't a clear indication as to why results on Antarctic sea ice would be relavent here.

---

## Referee Comment (RC2) · Anonymous Referee #2 · 24 Apr 2018

The current study aims to assess the ability of CMIP5 models to simulate past (1979-2014) evolution of sea ice. The authors construct a score that is supposed to capture the quality of the simulations. Based on this score they group the models and show that there are significant differences between the groups. Finally, they attempt to provide a physical explanation for the differences in performance. Unfortunately, the study is fundamentally flawed on various levels, and I cannot support the publication of this manuscript.

1) The score that is constructed by the authors (equation 4) is highly arbitrary and subjective. Additional weight is put on the Barents-Kara (BK) Seas. While a detailed assessment of the simulation quality of BK sea ice itself may be of interest, its combination with the simulation quality of sea ice outside the BK Seas makes the index

ambiguous and hard to interpret

2) The score combines a) a part that is related to the simulation of the trend in sea ice area and b) a part that is related to detrended sea ice concentration, based a a degree of agreement metric. Most of the detrended sea ice concentration is due to random year-to-year variations in sea ice concentration, which the CMIP5 models are not expected to capture (as they have not been initialized with observations). Therefore, it seems inappropriate to base the score on detrended sea ice concentration.

3) The HadISST1 data set used to score the model simulations is known to underestimate sea ice trends and is hence unsuitable. It would be better to use HadISST2 data.

4) The authors argue that some of the score differences can be explained by the stratospheric ozone data set used in the models. However, this is very hard to believe as there is no evidence in previous literature that stratospheric ozone variations and trends have a significant impact on Arctic sea ice. There is a lot of discussion on whether Antarctic sea ice is impacted by the Antarctic stratospheric ozone, but the potential impact of Arctic stratospheric ozone variations and trends (which are smaller than in the Antarctic) on Arctic sea ice (which has completely different drivers than Antarctic sea ice) is a completely different issue.

5) The authors group the models according to their score, and than show that the skill is different in different groups (e.g. Fig. 4 and Fig. 5). These are trivial results and the direct result of the way the score is constructed

6) The presentation of the paper is poor as it contains many grammatical errors.

---

## Author Comment (AC1) · 20 Jun 2018

Response to reviewer comments on "Assessment of Arctic sea ice simulations in CMIP5 models":
We would like to first thank the anonymous reviewers for their very helpful reviews on the paper. We hope that we have satisfactorily responded to all comments, which can be found here below. We apologize in advance for any repetition in our answers below: we wanted to respond to comments point-by-point.

**Anonymous Referee #1**

The study applies a new metric to assess the ability of CMIP5 models to accurately simulate Arctic sea ice evolution. Based on their results, they suggest that model performance strongly depends on ozone datasets and how well the Barents and Kara seas are simulated.
Originality: I very much like the idea of using physical mechanisms to examine the accuracy of Arctic sea ice evolution in climate models.
Scientific Quality: The authors, motivated by a number of studies outlined in the introduction, choose a metric to group the models based on how well they simulate the Barents and Kara seas. However, there choice of metric seemed arbitrary at best, and there was little analysis done to validate this method. For example, perhaps it could be useful to examine the relationship between BK sea ice trends vs pan-Arctic sea ice trends. This might help to convince readers of the method, and therefore make the rest of the analysis more convincing.

Response: Thank you for your insightful suggestions and comments! We choose the metric based on our understandings of asymmetric sea ice variability within the Arctic regions and the significance of BK sea ice variability on the Arctic climate change. BK sea ice variability is most remarkable within the Arctic region both in the interannual and in the multi-decadal time scales. Following your suggestions, we estimated the connection between BK sea ice trends and Pan-Arctic sea ice trends. Firstly, the sea ice concentrations are regressed onto the BK SIA index. Then we multiplied the regression coefficients with the linear trend of BK SIA to obtain Arctic sea ice trends that is congruence to BK sea ice trend. Figure 1 shows the ratio of BK-related SIA trend to the total SIA trend in each grid. Most of the Arctic region exhibits the same decreasing trends associated with the BK sea ice retreat. Nevertheless, we recognized that the weight coefficients are somewhat arbitrary and subjective. We used this metric to emphasize the relative importance of different items qualitatively. This score, combining BK and other Arctic regions (exBK), is intended to provide a synthetic assess method for the model sea ice simulation.

[Figure]

Fig 1. The ratio of BK-related SIA trend to the total SIA trend in Arctic.

Additionally, I was a bit surprised to not see Stroeve and Notz 2015 referenced in this paper. Stroeve/Notz go into great detail regarding how well Arctic sea ice is simulated by CMIP5 models Significance: I really like the idea of this sort of analysis, but it was not clear to me how this method improves on earlier studies that examine how well sea ice is simulated in climate models (e.g. Stroeve and Notz 2015). For example, it could be really interesting to examine what is learned by using their metric versus just examining the pan-Arctic, as is often done in these sorts of analyses?

Response: Thanks for your reminding. Stroeve and Notz 2015 indeed reviewed the past and future sea ice evolution by comparing CMIP5 model simulations with the observations. They reported that a robust reduction of the uncertainty range of future sea-ice evolution remains difficult for models. Actually, our assessment is attempting to constrain the uncertainty range to some extent. Figure 2 displays the root mean square (RMS) errors of SIA seasonal cycle (a), standard deviation(b) and running trends(c) between the observation and ensemble mean of models, the order of which is arranged by our weighting score method. It is obvious that the RMS errors decrease rapidly with the more models included at the first stage. However, the ensemble members are not the more the better. Once the ensemble members are over 10, the RMS errors will either level off or ascend, with more models included. In the light of this RMS analysis, we adjusted our sorting threshold and grouped the first 10 models as the first category.

This part about how we constrained the uncertainty range of models will be added in the revised manuscript.

[Figure]

Fig 2. a) The root mean square error of SIA seasonal cycle between observation and first n models' ensemble mean (n=2-29) arranged by score. The red dash line refers to the root mean square error of SIA seasonal cycle between all 30 models' ensemble mean and observation. b) Same as a, but for standard deviation of SIA for each month of the year. c) Same as a, but for the running decadal linear trend.

Presentation: There were a number of grammatical errors and typos throughout. Further, the introduction was a bit long, unclear, and took up about a 1/3 of the paper. Also, there seemed to be several references to Antarctic sea ice papers, though this was entirely an analysis of the Arctic

sea ice. There wasn't a clear indication as to why results on Antarctic sea ice would be relavent here.

Response: Thank you! We will have all the grammatical errors and typos fix in the revision. The unrelated content in introduction will be deleted and some content will be arranged logically.

**Anonymous Referee #2**

The current study aims to assess the ability of CMIP5 models to simulate past (1979-2014) evolution of sea ice. The authors construct a score that is supposed to capture the quality of the simulations. Based on this score they group the models and show that there are significant differences between the groups. Finally, they attempt to provide a physical explanation for the differences in performance. Unfortunately, the study is fundamentally flawed on various levels, and I cannot support the publication of this manuscript.

1) The score that is constructed by the authors (equation 4) is highly arbitrary and subjective. Additional weight is put on the Barents-Kara (BK) Seas. While a detailed assessment of the simulation quality of BK sea ice itself may be of interest, its combination with the simulation quality of sea ice outside the BK Seas makes the index ambiguous and hard to interpret.

Response: Thank you for your suggestions! We recognized that the weight coefficients are somewhat arbitrary and subjective. But we choose the metric based on our understandings of asymmetric sea ice variability within the Arctic regions and the significance of BK sea ice variability on the Arctic climate change. BK sea ice variability is most remarkable within the Arctic region both in the interannual and in the multi-decadal time scales. Further statistical analysis also shows that the BK sea ice variability have a close linkage with the other Arctic regions (Fig. 1).

2) The score combines a) a part that is related to the simulation of the trend in sea ice area and b) a part that is related to detrended sea ice concentration, based a a degree of agreement metric. Most of the detrended sea ice concentration is due to random year-to-year variations in sea ice concentration, which the CMIP5 models are not expected to capture (as they have not been initialized with observations). Therefore, it seems inappropriate to base the score on detrended sea ice concentration.

Response: Although the sea ice concentration anomaly of models may not be well simulated in the CMIP5 models, it's a very important variable reflecting the multi-scale atmosphere-sea ice interaction. Furthermore, we do validate the method by investigating the factors other than the detrended anomalies, such as in the SIA season cycle, SIA climatology mean and SIA trend in all meridians. The significant discrepancies among the groups in these three factors fully demonstrated the rationality of this assessment framework.

3) The HadISST1 data set used to score the model simulations is known to underestimate sea ice trends and is hence unsuitable. It would be better to use HadISST2 data.

Response: Thank you for your suggestions! We compared the HadISST1 and HadISST2 dataset, in reference to NSIDC satellite sea ice data, as shown in Fig 3. The climatology of HadISST1 is highly consistent with satellite data. In contrast, HadISST2 SIE climatology is strikingly overestimated by almost 5 million square kilometers (Fig. 3a)! For the SIE anomalies (by subtracting the climatology), HadISST1 data indeed underestimated the sea ice trend since the late 1990s. But the HadISST2 obviously overestimated the trend. As our scoring system contains the assessment of climatology, HadISST1 dataset is more suitable than HadISST2.

[Figure]

Fig 3. a) Monthly Arctic sea ice extent (SIC > 15%) during 1979–2016 from NSIDC datasets (gray), HadISST1 (red) and HadISST2 (blue). b) The same as a, but for SIE anomalies, and the linear trend of NSIDC datasets, HadISST1 and HadISST2 are -0.43, -0.24 and -0.63 million km²/10yr.

4) The authors argue that some of the score differences can be explained by the stratospheric ozone data set used in the models. However, this is very hard to believe as there is no evidence in previous literature that stratospheric ozone variations and trends have a significant impact on Arctic sea ice. There is a lot of discussion on whether Antarctic sea ice is impacted by the

Antarctic stratospheric ozone, but the potential impact of Arctic stratospheric ozone variations and trends (which are smaller than in the Antarctic) on Arctic sea ice (which has completely different drivers than Antarctic sea ice) is a completely different issue.

Response: The relevant report about the direct impact of stratospheric ozone on Arctic sea ice is hard to be found. However, the linkage between the Arctic ozone loss, polar vortex and Arctic Oscillation index has been revealed in many previous studies (e.g. Zhang Y, et al., 2013). In the light of close interaction between the Arctic Oscillation and the Arctic sea ice cover, we inferred that the stratospheric ozone may be a potential factor to influence the models' performance on the sea ice simulation. The statistical analysis in our study further supported this inference.

5) 5) The authors group the models according to their score, and than show that the skill is different in different groups (e.g. Fig. 4 and Fig. 5). These are trivial results and the direct result of the way the score is constructed

Response: Thanks! We do further RMS analysis and show the distribution of skill in Fig. 2 to demonstrate the grouping criterion. It is obvious that the RMS errors decrease rapidly with the more models included at the first stage. However, the ensemble members are not the more the better. Once the ensemble members are over 10, the RMS errors will either level off or ascend, with more models included. In the light of this RMS analysis, we adjusted our sorting threshold and grouped the first 10 models as the first category. This part about how we constrained the uncertainty range of models will be added in the revised manuscript.

6) The presentation of the paper is poor as it contains many grammatical errors.
Response: Thanks for your suggestion! I will have all the grammatical errors and typos fixed in my next revision.

Reference
Zhng Y, Wang W H, zhang X Y, et al., Interannual variations of Arctic ozone and their relationship to the Polar Vortex[J], Journal of Remote Sensing, 2013, 17(3):527-540
Stroeve J, and Notz D. Insights on past and future sea-ice evolution from combining observations and models[J]. Global and Planetary Change, 2015, 135:119-132.